# VQ-CAD: COMPUTER-AIDED DESIGN MODEL GENERATION WITH VECTOR QUANTIZED DIFFUSION

## ABSTRACT

Computer-Aided Design (CAD) software remains a pivotal tool in modern engineering and manufacturing, driving the design of a diverse range of products. In this work, we introduce VQ-CAD, the first CAD generation model based on *Denoising Diffusion Probabilistic Models*. This model utilizes a vector quantized diffusion model, employing multiple hierarchical codebooks generated through VQ-VAE. This integration not only offers a novel perspective on CAD model generation but also achieves state-of-the-art performance in 3D CAD model creation in a fully automatic fashion. Our model is able to recognize and incorporate implicit design constraints by simply forgoing traditional data augmentation. Furthermore, by melding our approach with CLIP, we significantly simplify the existing design process, directly generate CAD command sequences from initial design concepts represented by *text* or *sketches*, capture design intentions, and ensure designs adhere to implicit constraints.

## 1 INTRODUCTION

Computer-Aided Design (CAD) techniques remain a cornerstone in today's engineering and manufacturing landscape, underpinning the design of everything from the most rudimentary household items to sophisticated aircraft (Ikubanni et al., 2022; Shahin, 2008; Camba et al., 2016). The power of CAD lies in its ability to offer an intricate design platform that melds efficiency with precision. Typically, designers employ a "sketch-and-extrude" technique (Oh et al., 2006; Ren et al., 2022; Li et al., 2023) for CAD model creation, starting with 2D curves to define contours and then transforming them into 3D shapes. The culmination of these stages results in detailed CAD models.

While "sketch-and-extrude" is intuitive and efficient, it inevitably has various limitations. In intricate CAD designs, discrepancies often manifest when parameters fail to align with the set of requirements. These deep-rooted issues are not only elusive but also require significant time and resources for rectification. Despite the progress that has been made in CAD model generation through deep learning techniques (Wu et al., 2021; Willis et al., 2021; Xu et al., 2022; Lambourne et al., 2022; Xu et al., 2023), there remains a conspicuous oversight in addressing the automation of error correction, which is a challenge central to our research. Studies like (Para et al., 2021; Ganin et al., 2021; Seff et al., 2021) probe into the display constraints intrinsic to sketch generation, however, their reliance predominantly leans towards explicit constraints. We delve deeper, analyzing from a theoretical perspective and validating through experiments, the potential reasons why prior works may generate sketches that violate these constraints. Our solution is straightforward yet highly effective, *i.e.*, we give up data augmentation during the transformer training stage. A hallmark of our approach is its adeptness at autonomously generating commands without resorting to any explicit constraints, ensuring natural conformity to core design principles.

Building on this foundation, we aim to use *Denoising Diffusion Probabilistic Models* (DDPM), which has been proven effective in many fields (Ho et al., 2020; Lugmayr et al., 2022), to enhance the quality of the generated results. However, applying diffusion to CAD commands directly presents challenges. Inspired by *Vector Quantized Diffusion* (VQ-Diffusion) (Gu et al., 2022) model, we incorporate vector quantization in conjunction with hierarchical representation (Xu et al., 2023) to facilitate discrete diffusion within the latent space. Furthermore, we have improved our system by leveraging *Contrastive Language-Image Pre-Training* (CLIP) (Radford et al., 2021). This integration permits a seamless transition from image sketches and text descriptions directly to CAD

sequences, ensuring a level of implicit regularity. By utilizing CLIP's capabilities, we not only open doors for a more intuitive design process, but also facilitate a robust mechanism for error detection and correction. This fusion of traditional CAD design with CLIP's advanced methodologies offers a glimpse into the future of CAD design, where systems do not just follow commands but actively learn, adapt, and even correct. This represents not just an advancement for design professionals, but also a meaningful progression in CAD research, transitioning from fixed manual constraints to a more adaptive, learning-focused design approach.

The main contributions are summarized as follows:

- We introduce VQ-CAD, a novel and effective approach for CAD model generation by utilizing VQ-Diffusion.

- Our model shows a unique ability to discern implicit design constraints, paving the way for potential automated error rectification mechanisms in CAD design.

- We further enhance the CAD model generation system by incorporating the strengths of CLIP, facilitating a seamless transition directly from image sketches and text descriptions to CAD sequences.

## 2 RELATED WORK

**Constructive Solid Geometry Learning.** Constructive Solid Geometry (CSG) is a technique for creating complex 3D models by applying Boolean operations on geometry primitives. Despite its wide applications in like mechanical manufacturing and architectural design, the process can be complex. Recently, deep learning techniques have been employed to study the process of generating 3D models using CSG. The first such method is CSGNet (Sharma et al., 2018), which employs reinforcement learning to identify CSG commands that minimize reconstruction error. UCSGNet (Kania et al., 2020) provided a larger solution space by performing multiple Boolean operations on the generated primitives. CSG-Stump (Ren et al., 2021) optimized the solution space, outputting CSG programs composed of unions of intersections of primitives or their complements. Similarly, CAPRI-Net and $D^2$CSG (Yu et al., 2022; 2023) transformed original quadratic surfaces into convex bodies through intersection and defined two types of shapes by uniting these bodies, with the final output being the difference between these shapes. These pioneering works have opened up new avenues for reconstructing generative commands for 3D shapes through deep learning methods.

**Sketch and Extrude CAD Generation.** In terms of 3D model reconstruction, Point2Cyl (Uy et al., 2022) is an innovative supervised network that effectively transformed 3D point clouds into a set of extrusion cylinders. ExtrudeNet (Ren et al., 2022) and SECAD-Net (Li et al., 2023) extended the CAPRI-Net (Yu et al., 2022) architecture by replacing convex bodies created by intersecting quadratic surfaces with extruded 2D sketches while maintaining the end-to-end differentiability. Sketch2CAD, CAD2Sketch, and Free2CAD (Li et al., 2020; 2022; Hähnlein et al., 2022) learned to convert between hand drawings and CAD commands. DeepCAD (Wu et al., 2021) is a forerunner in 3D model generation by translating shapes into sequences of CAD operations with a transformer-based generative network, which shows its efficacy in both shape auto encoding and random shape generation. SkexGen (Xu et al., 2022) deployed auto-regressive generative models and distinct transformer architectures for CAD construction sequences, allowing efficient design space exploration. It improved the user control and the production of diverse CAD models. Additionally, HNC-CAD (Xu et al., 2023) employed a hierarchical tree of neural codes for high-level design concepts, excelling in tasks like unconditional generation and enhancing conditional generation through code tree manipulations. Compared to these methods, we can generate 3D models that better satisfy implicit design constraints and our model has the ability to generate CAD construction sequences from prompts.

**Discrete Diffusion Models.** DDPM have been recognized for their effectiveness in generative modeling, particularly in image generation and restoration (Ho et al., 2020; Saharia et al., 2022; Lugmayr et al., 2022). The advent of Latent Diffusion Models (LDM) extended DDPM to the latent space, expanded the capabilities of DDPM into the latent space, yielding promising results across various domains, including high-resolution image synthesis and brain imaging (Vahdat et al., 2021; Rombach et al., 2022; Vahdat et al., 2022).

When confronted with discrete data like text, DDPM faces more challenges. The foundation of diffusion models for discrete state spaces can be traced back to the work by Sohl-Dickstein et al. (2015), which introduced a diffusion process for binary random variables. This concept was later expanded to categorical random variables with uniform transition probabilities by Hoogeboom et al. (2021). Building on these concepts, D3PM (Austin et al., 2021) devised a comprehensive framework for diffusion processes involving categorical random variables, which have been applied in tasks such as text-to-sound generation (Yang et al., 2023). Inspired by this, VQ-Diffusion Gu et al. (2022), a blend of VQ-VAE (Van Den Oord et al., 2017) and conditional DDPM, has shown potential in text-to-image generation, with its "mask-and-replace" approach effectively reducing prediction errors. In addition to text-to-image generation, Inoue et al. (2023) has validated the effectiveness of VQ-Diffusion in the field of layout generation.

**Text-to-Shape Generation.** The recent surge in technological advancements has brought the generation of text-to-shape to the forefront. However, the lack of sufficient text-to-shape datasets poses a significant challenge, confining the applications of supervised generation techniques to limited categories. CLIP-Draw (Frans et al., 2022) circumvented this limitation by marrying differentiable rendering with CLIP for the synthesis of text-to-drawn. Similarly, numerous studies (Jain et al., 2022; Mohammad Khalid et al., 2022; Poole et al., 2022; Wang et al., 2023) have leveraged the image and text embeddings provided by CLIP (Radford et al., 2021), utilizing rendering techniques to generate 3D models. Deviation from this trajectory, CLIP-Forge and CLIP-Sculptor (Sanghi et al., 2022; 2023a) utilize CLIP for conditional generation in the latent space, advancing text-driven zero-shot 3D generation. Similarly, Sketch-a-Shape (Sanghi et al., 2023b) employs a pre-trained image encoder and a masked transformer to transform sketches into various 3D shape representations.

Our approach distinguishes itself by seamlessly intertwining *CLIP, VQ-Diffusion and hierarchical code tree*. This integration facilitates the conversion of images, sketches, and textual directives into CAD command sequences, satisfied with the inherent regularization constraints. This emerging field holds huge potential, especially in areas such as design, art, and content generation. Moreover, the ability to automatically extract CAD instructions from mere textual descriptions or rudimentary sketches can drastically enhance operational efficiency.

## 3 METHODOLOGY

### 3.1 CAD SEQUENCE REPRESENTATION

In sketch-and-extrude model, designs emerge from multiple extruded sketches. Each sketch is assembled from faces, which are organized by loops. These loops encompass curves, *e.g.*, lines, arcs, or circles, which are composed of 2, 3, or 4 points, respectively. Besides the representation of sketch-and-extrude sequences, we also adopt a hierarchical structure containing three main levels: *loops, profiles, and solids* (Xu et al., 2023). A more elaborate discourse on this can be found in the *Appendix*.

### 3.2 VQ-CAD

When directly conducting the diffusion process on CAD sequences, we found that this naive method produces limited generation performance. A possible reason is that the length of the CAD data sequence is too long to directly model. Inspired by latent diffusion models, we attempted to conduct diffusion in the latent space. With the help of the hierarchical representation, CAD command sequences are converted into a code tree, where each code is looked up from codebooks of VQ-VAE trained specifically for loops, profiles, and solids. This approach enables us to perform discrete diffusion in this discrete latent space. Figure 1 outlines our proposed VQ-CAD method with a three-stage training process.

In the codebook extraction phase, we use the same version of VQ-VAE as HNC-CAD with Masked Autoencoder (He et al., 2022). While developing the code tree decoder, we found that data augmentation disrupts the implicit regularity inherent in the data. This observation has significant implications for our method, which will be further discussed in Section 4.2. Next, we briefly introduce the VQ-Diffusion module.

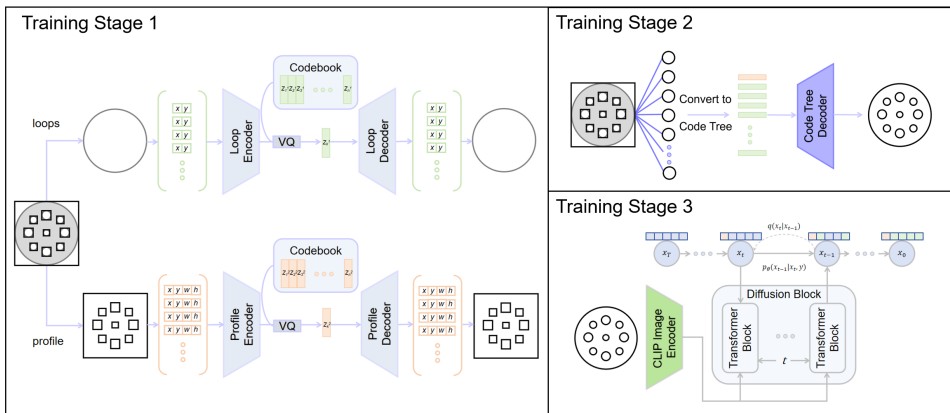

Figure 1: The illustration of the three-stage training process for transforming 2D sketches into CAD models of the proposed VQ-CAD method. In the first stage, separate VQ-VAEs are trained for "loop" and "profile" elements, generating individual codebook encodings for each object. Then, sketches are converted into corresponding code tree, which are then paired with their respective CAD commands to train the code-tree decoder. Finally, a frozen CLIP image encoder is employed to encode the sketch images, providing conditional guidance for diffusion.

### 3.2.1 DISCRETE DIFFUSION MODELS

To carry out a discrete diffusion process on the code tree, we adopt the approach of VQ-Diffusion. Specifically, for discrete random variables that cover $K$ categories, denoted as $x_t, x_{t-1} \in \{1, \ldots, K\}$, the forward transition probabilities can be expressed as:

$$[\boldsymbol{Q}_t]_{ji} = q(x_t = j | x_{t-1} = i). \tag{1}$$

Given the one-hot representation of $x$, denoted as row vector $\boldsymbol{x}$, the relationship is:

$$q(\boldsymbol{x}_t | \boldsymbol{x}_{t-1}) = \boldsymbol{x}_t^\top \boldsymbol{Q}_t \boldsymbol{x}_{t-1}. \tag{2}$$

Starting from $\boldsymbol{x}_0$, the $t$-step marginal and the posterior at time $t-1$ are:

$$q\left(\boldsymbol{x}_t | \boldsymbol{x}_0\right) = \boldsymbol{x}_t^\top \overline{\boldsymbol{Q}}_t \boldsymbol{x}_0, \quad \text{where } \overline{\boldsymbol{Q}}_t = \boldsymbol{Q}_1 \boldsymbol{Q}_2 \ldots \boldsymbol{Q}_t. \tag{3}$$

$$
\begin{aligned}
q\left(\boldsymbol{x}_{t-1} | \boldsymbol{x}_t, \boldsymbol{x}_0\right) &= \frac{q\left(\boldsymbol{x}_t | \boldsymbol{x}_{t-1}, \boldsymbol{x}_0\right) q\left(\boldsymbol{x}_{t-1} | \boldsymbol{x}_0\right)}{q\left(\boldsymbol{x}_t | \boldsymbol{x}_0\right)} \\
&= \frac{\left(\boldsymbol{x}_t^\top \mathbf{Q}_t \boldsymbol{x}_{t-1}\right)\left(\boldsymbol{x}_{t-1}^\top \overline{\mathbf{Q}}_{t-1} \boldsymbol{x}_0\right)}{\boldsymbol{x}_t^\top \overline{\mathbf{Q}}_t \boldsymbol{x}_0}.
\end{aligned}
\tag{4}
$$

In the reverse denoising process, a straightforward approach is to use a bidirectional transformer encoder blocks to fit the distribution $p_\theta(\boldsymbol{x}_{t-1} | \boldsymbol{x}_t)$. However, similar to the methods used in prior work (Sohl-Dickstein et al., 2015; Hoogeboom et al., 2021; Austin et al., 2021; Gu et al., 2022), we integrate an additional neural network $\tilde{p}_\theta(\tilde{\boldsymbol{x}}_0 | \boldsymbol{x}_t)$. By summing over potential $\tilde{x}_0$ values, it can be transformed into a one-step reverse denoising process, yielding the following representation:

$$p_\theta(\boldsymbol{x}_{t-1} | \boldsymbol{x}_t) = \frac{\sum_{\tilde{x}_0} \tilde{p}_\theta(\tilde{\boldsymbol{x}}_0 | \boldsymbol{x}_t) \cdot q(\boldsymbol{x}_{t-1} | \boldsymbol{x}_t, \tilde{\boldsymbol{x}}_0)}{\sum_{\boldsymbol{x}_{t-1}} \sum_{\tilde{x}_0} \tilde{p}_\theta(\tilde{\boldsymbol{x}}_0 | \boldsymbol{x}_t) \cdot q(\boldsymbol{x}_{t-1} | \boldsymbol{x}_t, \tilde{\boldsymbol{x}}_0)} \propto \sum_{\tilde{x}_0} \tilde{p}_\theta(\tilde{\boldsymbol{x}}_0 | \boldsymbol{x}_t) \cdot q(\boldsymbol{x}_{t-1} | \boldsymbol{x}_t, \tilde{\boldsymbol{x}}_0) \tag{5}$$

The loss function, which includes both the conventional variational lower bound $\mathcal{L}_{\text{vlb}}$ and an auxiliary denoising objective, is given by:

$$\mathcal{L}_\lambda = \mathcal{L}_{\text{vlb}} + \lambda \mathbb{E}_{\substack{\boldsymbol{x}_t \sim q(\boldsymbol{x}_t | \boldsymbol{x}_0) \\ \boldsymbol{x}_0 \sim q(\boldsymbol{x}_0)}} \left[ -\log \tilde{p}_\theta(\boldsymbol{x}_0 | \boldsymbol{x}_t) \right], \tag{6}$$

where $\lambda$ is a balancing hyper-parameter. VQ-Diffusion introduces an enhancement to $\mathbf{Q}_t$ via the mask-and-replace strategy. The transition matrix $\mathbf{Q}_t$ is:

$$\mathbf{Q}_t = \begin{bmatrix} \alpha_t + \beta_t & \beta_t & \cdots & \beta_t & 0 \\ \beta_t & \alpha_t + \beta_t & \cdots & \beta_t & 0 \\ \vdots & \vdots & \ddots & \beta_t & 0 \\ \beta_t & \beta_t & \beta_t & \alpha_t + \beta_t & 0 \\ \gamma_t & \gamma_t & \gamma_t & \gamma_t & 1 \end{bmatrix}, \tag{7}$$

where $\alpha_t$, $\beta_t$ and $\gamma_t$ are designed such that $z_t$ converges to the [MASK] token as $t$ grows. During testing, we start from $\boldsymbol{x}_t$ with [MASK] tokens and iteratively sample $\boldsymbol{x}_{t-1}$ from $p_\theta(\boldsymbol{x}_{t-1}|\boldsymbol{x}_t)$.

### 3.2.2 UNCONDITIONAL GENERATION

In the task of unconditional CAD sequence generation, the initial step involves compressing loops and profiles within each sequence into codebook using VQ-VAE, as shown in the training stage 1 in Figure 1. Similar to the training stage 2, a code tree decoder is trained. Each sequence is then represented by code tree, enabling a diffusion process that begins from the mask within the sequence. During the training of this diffusion model, no conditions are applied. In the inference phase, the diffusion model is first used to generate a code tree, which is then decoded by the code tree decoder to produce a complete command sequence.

### 3.2.3 CONDITIONAL GENERATION

To facilitate conditional generation, we channel conditions to the model via the cross-attention of a transformer decoder. To streamline the training process, we limit our conditional generation experiments to sketches. This approach can be expanded to 3D models by training from photographs taken from different angles, akin to methodologies in CLIP-Sculptor and CLIP-Forge. As part of our approach, we utilize a pre-trained and frozen version of CLIP as our encoder.

More specifically, as shown in Figure 1, during the training phase, our dataset comprises solely of command sequences. We first generate a code tree from the initial sequence, mirroring the approach for unconditional generation. Concurrently, we employ a parser to convert the initial sequence into an image. During the training, we leverage a frozen CLIP image encoder to encode the image and derive its latent vector. This

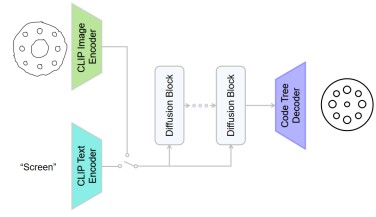

Figure 2: Inference schematic. Utilizing CLIP's text or image encoder to encode conditional information for guiding the diffusion process.

latent vector serves as the condition, with the code tree as the target, to train VQ-Diffusion. Given the robust encoding capabilities of CLIP's image encoder, our inference phase can handle not only correctly parsed images but also hand-drawn sketches, allowing for zero-shot predictions. Moreover, thanks to CLIP's contrastive learning, it can encode semantically similar images and texts into corresponding latent vectors. Through experiments, we discover that to some extent, we can employ CLIP's text encoder for text-driven CAD command synthesis, as depicted in Figure 2. Inspired by the prompt engineering (Radford et al., 2021), we further enhance the text input. Specifically, for each template in our template list, we fill in the desired object to be generated, creating a set of enriched text prompts. Each of these prompts is then fed into the text encoder separately, yielding a series of feature vectors. We compute the average of these feature vectors to obtain a consolidated feature representation for the desired object. By feeding this refined feature representation into the diffusion transformer block, we significantly boost the efficacy of text-guided CAD sequence generation process.

## 4 EXPERIMENTAL RESULTS

### 4.1 IMPLEMENTATION DETAILS

**Datasets.** In our study, we utilized the expansive DeepCAD dataset (Wu et al., 2021) that is annotated with sketch-and-extrude models. This dataset encompasses 178,238 such models, divided into

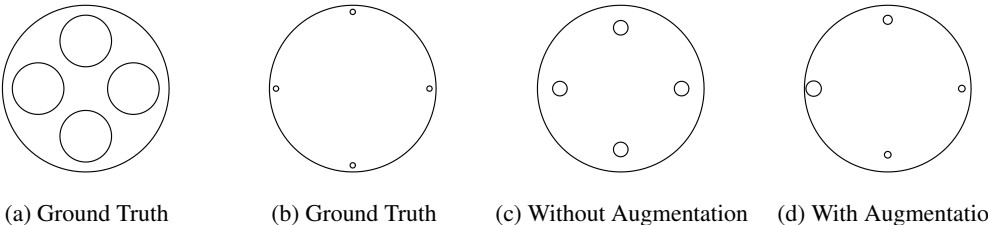

(a) Ground Truth      (b) Ground Truth      (c) Without Augmentation      (d) With Augmentation

Figure 3: Sketch examples: (a)&(b) are random training set sketches. (c) A symmetry-preserving sketch generated without data augmentation. (d) A sketch from training with data augmentation, losing the inner circle symmetry.

90% for training, 5% for validation, and the remaining 5% for testing. Drawing from methodologies in prior approaches (Xu et al., 2023; 2022; Willis et al., 2021), we identified and omitted duplicated models in the training subset. After delineating the hierarchical attributes for loop, profile, and solid, we proceeded to eliminate repetitive properties at each tier. Each coordinate was converted into a 6-bit number via quantization. Furthermore, our training was limited to CAD models that conformed to specific criteria: a maximum of 5 solids, up to 20 loops for every profile, no more than 60 curves for every loop, and an upper limit of 200 commands within the sketch-and-extrude sequence. Post this stringent duplication and filtration process, our training dataset comprised 102,114 solids, 60,584 profiles, and 150,158 loops for the purpose of codebook learning. Additionally, it contained 124,451 sketch-and-extrude sequences tailored for CAD model creation. For conditional generation, we adopted the approach from SkexGen (Xu et al., 2022) and HNC-CAD (Xu et al., 2023), and extracted sketches directly from DeepCAD dataset. Post the elimination of duplicates, our training was based on a total of 99,650 sketches. To demonstrate the capability of our conditional generation model, we utilized the test set to extract sketches for testing, resulting in a total of 4,842 sketches.

**Optimization and Configuration.** Models are optimized using an Nvidia A100 GPU. The Transformer backbone employs pre-layer normalization and consists of 4 layers, each with 8 attention heads. The input size for the embedding layer is set to 256, and the feed-forward network scales to a dimension of 512. During training, a dropout rate of 0.1 is applied to prevent the omission of features. We adopt a VQ-VAE codebook similar to the one utilized in HNC-CAD (Xu et al., 2023). For conditioned generation, we leverage the ViT-B/32 model from CLIP (Radford et al., 2021), based on the Vision Transformer architecture (Dosovitskiy et al., 2020), and freeze it to serve as our text and image encoder. The parameter $\lambda$ in Equation 6 is set to 0.1, and the number of diffusion timesteps $T$ is 100. For optimization, the AdamW optimizer (Loshchilov & Hutter, 2018) is used with a learning rate of $5.0 \times 10^{-4}$, $\beta_1 = 0.9$, and $\beta_2 = 0.98$.

**Evaluations.** To measure the capability of generating CAD models, we convert them to point clouds, and then use the Coverage (COV), Jensen-Shannon Divergence (JSD), and Maximum Mean Discrepancy (MMD) for comparision (Achlioptas et al., 2018; Wu et al., 2021; Xu et al., 2021; 2023). To assess novelty, we further compute Novel and Unique values between the generated command sequences and the training set. (Wu et al., 2021; Xu et al., 2021; 2023). To evaluate the regularity of generated sketches, we introduce a symmetry metric, which assesses symmetry through horizontal and vertical flipping based on the sketch's centroid coordinates. The symmetry metric is given by:

$$\text{Symmetry}(I) = \frac{1}{N} \sum_{i=1}^{h} \sum_{j=1}^{w} |I(i,j) - I(i, 2O_y - j)| + \frac{1}{N} \sum_{i=1}^{h} \sum_{j=1}^{w} |I(i,j) - I(2O_x - i, j)| \quad (8)$$

Here, $w$ and $h$ are the width and height of the image, respectively, $N$ is the total number of pixels in the image, and $O$ represents the centroid with coordinates $(O_x, O_y)$. Further details are provided in the *Appendix*.

## 4.2 DATA AUGMENTATION AND ITS EFFECTS ON IMPLICIT STRUCTURE

In the realm of CAD sequence generation, previous works such as SkexGen and HNC-CAD have employed data augmentation techniques to enhance model robustness and performance. Specifically,

they introduced noise to the input of transformer decoders, whose objective remains the generation of correct sequences even in the presence of noise.

However, upon examining the generated sequences from HNC-CAD, anomalies became evident, *i.e.*,certain generated sequences exhibited characteristics that defied CAD logic, such as non-vertical sides of rectangles or asymmetrical objects that should ideally be symmetrical, as demonstrated in the following experiments. A deeper dive into this phenomenon suggested that the employed data augmentation might inadvertently disturb the intrinsic symmetries in the data distribution.

Many tasks, especially those involving sequence generation, involve data where elements have inherent relationships or dependencies. Data augmentation, like noise introduction, is commonly employed to improve model generalizability. Yet, this can sometimes have unintended ramifications on the data's implicit structure.

More specifically, consider a rudimentary dataset containing sequences $[0, 0]$ and $[1, 1]$. Let the first position in the sequence be $X$ and the second be $Y$. Here, $Y$ is inextricably linked to $X$. The conditional probabilities for this setup are:

$$P(Y = 0|X = 0) = P(Y = 1|X = 1) = 1, \quad P(Y = 1|X = 0) = P(Y = 0|X = 1) = 0.$$

Assuming a data augmentation strategy introduces noise that alters the first position value with the probability equating to 0.5. This leads to potential sequence variations. Subsequent to this modification, the conditional probabilities evolve:

$$P(Y = 0|X = 0) = P(Y = 1|X = 0) = P(Y = 1|X = 1) = P(Y = 0|X = 1) = 0.5.$$

The advent of noise alters the model's optimization objectives. From comprehending deterministic relationships in the original data, the model gravitates towards learning a uniform conditional probability distribution. This metamorphosis underscores the nuances and pivotal considerations imperative when wielding data augmentations, as they can unintentionally modify a dataset's intrinsic fabric.

To validate the correctness of our analysis, we selected a sketch sharing the same code tree as a mini-dataset. This dataset comprises 27 images, all constructed from one large outer circle and four smaller inner circles. Theoretically, this scenario mirrors the previously discussed example. As shown in Figure 3, models trained with data augmentation produced instances where the sizes of the four inner circles vary, disrupting the desired symmetry. In contrast, models trained without data augmentation better preserved the symmetry of the four inner circles, validating our observation.

### 4.3 UNCONDITIONAL GENERATION

To demonstrate the capabilities of our diffusion module, we conducted experiments on the well-studied 3D generation task and compared it with state-of-the-art methods. Specifically, we still adopted the learning process depicted in the diagram, but did not use conditions for supervision. During generation, we performed random diffusion.

As illustrated in Table 1, our method surpasses the baselines in two evaluation metrics: COV and JSD, demonstrating significant improvements in quality and diversity. In terms of Unique and Novel scores, our performance either matches or closely approaches the state of the art, indicating that VQ-Diffusion has a better representation of the sample distribution and can produce more diverse results. Our generations exhibit enhanced diversity and novelty similar to SkexGen, yet with substantially improved COV and JSD scores. When pitted against HNC-CAD, not only do our COV and JSD

Table 1: Quantitative evaluations on the CAD generation task. We adopt COV, MMD, JSD, and Percentage Scores for Unique and Novel for result assessment. **Bold** fonts indicate the best generator.

| **Method** | COV % ↑ | MMD ↓ | JSD ↓ | Novel % ↑ | Unique % ↑ |
|---|---|---|---|---|---|
| DeepCAD (Wu et al., 2021) | 80.62 | 1.10 | 3.29 | 91.7 | 85.8 |
| SkexGen (Xu et al., 2022) | 84.74 | 1.02 | 0.90 | **99.1** | 99.8 |
| HNC-CAD (Xu et al., 2023) | 87.73 | **0.96** | 0.68 | 93.9 | 99.7 |
| **Ours** | **88.11** | 1.05 | **0.64** | 98.0 | **99.9** |

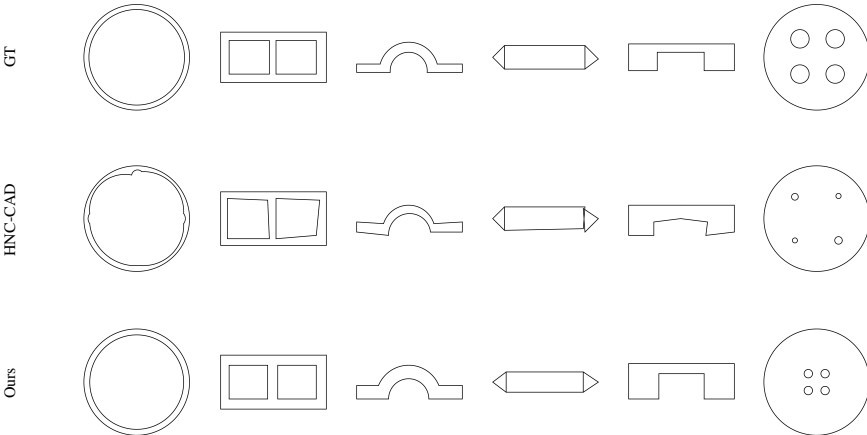

Figure 4: Picture-to-Command result comparison: The first row displays input images. The second and third rows show the reconstruction results of HNC-CAD (Xu et al., 2023) and ours, respectively.

scores fare better, but we also register significantly higher Novel score. Although our MMD lags slightly behind SkexGen, this is attributed to the greater size diversity in our generations, leading to some samples having a larger minimum distance from the test set. The JSD metric underscores that the point clouds we generated align closer to the ground truth when viewed from the vantage point of probability distribution.

## 4.4 CONDITIONAL GENERATION

**Picture to CAD.** To demonstrate the capability of our algorithm, we modified the HNC-CAD algorithm to use the same condition for training.

The testing results are shown in Table 2. We use Symmetry as the error metric. It can be observed that our algorithm has advantages over the competing method HNC-CAD. As shown in Figure 4, our algorithm can produce CAD sequences that adhere more closely to the implicit rules, which is more in line with the rules of CAD design.

Table 2: Quantitative comparisons on the *Symmetry* metric.

|  | Symmetry ↓ |
| --- | --- |
| GT | 0.03907 |
| HNC-CAD | 0.04017 |
| **Ours** | **0.03921** |

**Zero-Shot CAD Generation.** In the domain of text-to-image generation, works such as VQ-Diffusion and Latent Diffusion have demonstrated that diffusion models possess certain advantages over transformer models. Given this, we further explored this phenomenon in the context of CAD sequence generation. In our model, we replaced the diffusion mechanism with a transformer module, maintaining all other components unchanged. Considering the limited size of the CAD sequence dataset, which cannot generate meaningful CAD sequences for all texts, we selected specific texts that can produce meaningful CAD sequences for comparative experiments. Building on this foundation and incorporating prompt engineering, as shown in Figure 5, observations revealed that for fundamental concepts like circle and rectangle, transformers can synthesize more precise and succinct CAD command sequences. However, when dealing with complex concepts such as volcano and rocket, transformer fails to deliver the expected output.

We also endeavored to generate CAD sequences from sketches using models trained on regular images. As depicted in Figure 6, even with sketches that have substantial random noise, the model is still capable of generating reasonably coherent CAD sequences to some extent.

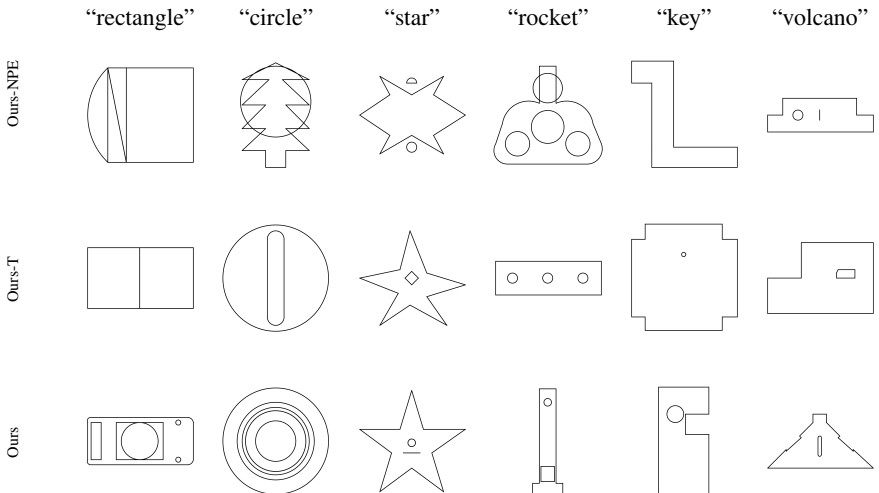

Figure 5: Text-to-Command generation results of different methods: Given the top text as a condition, each row of images below represents the output of various models based on the provided condition. In this comparison, "Ours-NPE" denotes "Ours with No Prompt Engineering," and "Ours-T" refers to "Ours with Transformer."

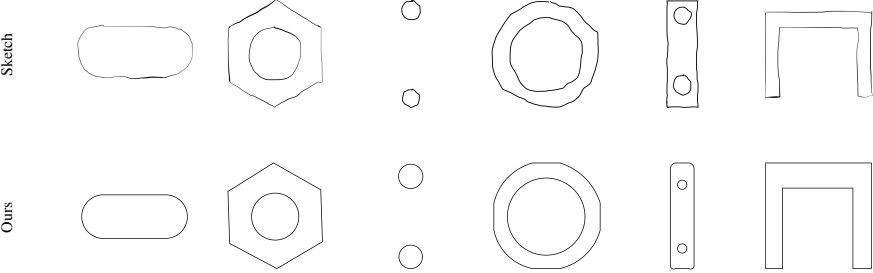

Figure 6: Hand-drawn Sketch-to-Command generation results: The first row displays hand-drawn sketches, the second row shows our generation results.

## 5 LIMITATIONS AND FUTURE WORK

We recognize that while our work is novel and effective, it bears certain limitations. The constrained size and lack of diversity in the dataset impede our capabilities in zero-shot generation. To mitigate this, future research could concentrate on amassing a more extensive set of CAD command data. Moreover, we can enhance the geometric accuracy of our model by incorporating relevant loss functions, which presents further opportunities for refinement in subsequent studies. Our approach can be expanded to 3D models by training from photographs taken from different angles, enabling the generation of CAD sequences from text and hand-drawn sketches seamlessly.

## 6 CONCLUSIONS

In this work, we introduce an approach to CAD model generation by leveraging the capabilities of VQ-Diffusion. Our methodology distinguishes itself through its inherent ability to discern implicit design constraints, suggesting potential automated mechanisms for error correction in CAD designs. By integrating the robust features of CLIP, we unveil an enhanced system for generating CAD models. This synergistic approach facilitates a seamless transition from image sketches and textual annotations directly to CAD sequences, achieving a level of design regularity. This methodology signifies not only an advancement for practitioners in design but also inaugurates new avenues in CAD research, transitioning from a command-centric to a more agile, learning-focused design paradigm.

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
