# OpenReview forum: "VQ-CAD: Computer-Aided Design Model Generation with Vector Quantized Diffusion"
_ICLR.cc/2024/Conference — ICLR 2024 Conference Withdrawn Submission_

### Official Review · Reviewer_4YyQ · 2023-10-26

**Soundness:** 4 excellent
**Presentation:** 4 excellent
**Contribution:** 2 fair
**Rating:** 6
**Confidence:** 5

**Summary:**

This paper proposed a sketch-and-extrude CAD sequence generation pipeline based on VQ-Diffusion and hierarchical neural code tree. The encoded code tree index from VQ-VAE is used to train a VQ-Diffusion model, which was applied to various conditional generation tasks such as text to CAD, sketch to CAD. Evaluations show that the DM version outperforms the autoregressive version of HNC-CAD. And the overall pipeline itself can be treated as a uniform model for controllable CAD generation from different modalities.

**Strengths:**

The paper is well-written and easy to follow. Motivation for building a diffusion-based model for CAD generation is strong and well-aligned with the current trend in AIGC. Significant work was done to convert HNC-CAD and get it to work with VQ-Diffusion. Authors demonstrate the strong performance and generalization of their diffusion model across multiple CAD generation tasks, which was previously not shown to be solved by a uniform pipeline. This could set a standard for 3D CAD generation (e.g. like the stable diffusion for 2D image), which would be very impactful for the CAD community.

**Weaknesses:**

The proposed method is mostly based on two existing works (1) HNC-CAD for encoding the CAD design intent as a neural code tree, and (2) VQ-Diffusion for generating the code indices. I appreciate the effort that authors took to modify and piece together the two modules, but the algorithm contribution seems somewhat incremental in this case. With that being said, I am not fully against such a paper as stable diffusion is one of these in terms of algorithms. But I do have concern regarding the sketch-and-extrude DeepCAD dataset used to train the network. Unlike stable diffusion which is trained on large-scale dataset, DeepCAD is relatively small and excludes many useful operations such as chamfer or sweep. The text that was used for text to cad was also very simple and does not reflect the actual working conditions for designers. The question of whether those results can be used to support this pipeline as a general and effective approach for CAD generation remains questionable. So purely from a algorithm perspective the paper can only be rated as borderline.

**Questions:**

1) Stable diffusion has both VQ and KL versions. For VQ, networks just denoise to the code vector. That version was already proven to work very well. I wonder what motivates authors to use VQ-Diffusion with discrete translation matrix instead. From my understanding, code indices carry far fewer information than the code vector.
2)  What is the success rate for generating a valid CAD model? Usually for SkexGen and DeepCAD that number is not very high.  I wonder if diffusion is better at figuring out the correct alignment and constraint. The visual quality from the results seem to suggest so, but an actual score benchmark will be better for comparison.
3) Currently the model is not trained on large-scale dataset and the use of CLIP pretraining is rather limited. I think the paper will be much stronger if it demonstrates some results like in ReparamCAD. I wonder if authors can provide clarification on how their approach can significantly simplify the existing design process with the help of CLIP or other components, as opposed to showing better results on existing task settings because it combines two very powerful algorithms.

---

### Official Review · Reviewer_yDgr · 2023-10-30

**Soundness:** 1 poor
**Presentation:** 1 poor
**Contribution:** 2 fair
**Rating:** 3
**Confidence:** 3

**Summary:**

This paper presents a method for CAD model generation. This approach utilizes a hierarchical structure comprising three primary levels: loops, profiles, and solids to delineate a CAD model. Initially, separate VQ-VAEs are trained for both the "loop" and "profile" levels. Subsequently, a VQ diffusion model is trained to produce a code tree. To incorporate image or text conditioning, the method leverages CLIP to extract relevant features from images or texts. These features are then integrated into the training of the VQ diffusion.

**Strengths:**

The method employs VQ Diffusion for CAD model generation.

**Weaknesses:**

1. The structure and presentation of the paper could be significantly improved, as I found it challenging to follow the paper and comprehend its method. For example, I spent considerable time attempting to grasp the basic points:

(a) The definition of "loops", "profiles", and "solids" are unclear. How is the code tree represented, and what is its relationship with the command sequences? How can they be interconverted? Providing concrete examples would greatly aid comprehension.

(b) The input and output of the VQ-VAE are not clearly stated.

(c) How is a sketch translated into the code tree is unclear. The term "code tree decoder" also lacks clarity — what are its inputs and outputs?

(d) How does VQ diffusion is integrated into the pipeline is unclear. Also, how is the condition added into the diffusion block?

(e) You mentioned "perform discrete diffusion in this discrete latent space". However, the term "discrete latent space" is unclear and need clarification.

2. The paper dedicates substantial space to introducing VQ Diffusion, but many of the symbols and terms (e.g., Q_t, z_t, mask token) are still not defined or explained. Moreover, rather than providing an exhaustive introduction of the existing method of VQ diffusion, the authors should focus on elucidating how VQ diffusion is integrated into their pipeline.

3. The paper lacks technical novelty, with the application of VQ diffusion appearing straightforward and lacking in specific or insightful design. Additionally,  the observation that data augmentation might negatively impact performance also lacks depth.

4. I am skeptical that CLIP-derived text or image features can encapsulate the fine-grained details necessary to serve as conditions for guiding CAD model generation.

**Questions:**

Please see weaknesses.

---

### Official Review · Reviewer_AUuC · 2023-11-01

**Soundness:** 2 fair
**Presentation:** 2 fair
**Contribution:** 1 poor
**Rating:** 3
**Confidence:** 5

**Summary:**

This paper proposes VQ-CAD, a latent diffusion model for generating sketch-extrude CAD sequences that can be converted into editable parametric CAD models. Sketch constraints are an important part of parametric CAD, and rather than explicitly generating constraints, this paper implicitly learns the constraints following the closely related work, HNC-CAD. The effect of data augmentation used in previous works is analyzed . Finally, CLIP is integrated into the framework as a means to conditioning the generation of CAD sequences with images or text.

**Strengths:**

The main strength of the proposed method is its performance compared to previous work. It does perform slightly better than HNC-CAD/SkexGen in the unconditional generation task. The paper is well written, and easy to follow. Comparisons include all the recent sketch-extrude generative models. The empirical analysis on the effect of data augmentation is useful.

**Weaknesses:**

The main weakness of the paper is its limited novelty and limited exploration of applications.

The paper is heavily based on HNC-CAD (including the code-tree representation, most of the architecture, training schemes, etc.) with the only difference being the usage of a VQ-Diffusion model for generating the code trees rather than an autoregressive Transformer.

While using CLIP to condition CAD sequence prediction is interesting, the application was not deeply explored, and hence the technical contribution is limited from a research perspective. The image based conditioning works, but most of the qualitative results actually show that the conditioning is not being strongly enforced. Text to CAD generation does not appear to work well at all, presumably due to the domain gap between CLIP's training data and CAD data, as well as the abstract nature of the command sequence compared to other spatial representations like voxels or SDFs used in previous image/text-to-3D works. These problems were not sufficiently explored in the paper.

**Questions:**

- In Table 4 of the appendix, row 1, the novelty and unique metrics are low. But these metrics dependent on sampling hyperparameters such as the temperature, and the "p" value used in the top-p sampling used in the Transformer. Do these numbers change significantly when these hyperparameters are tweaked?
- How do existing latent generative models such as MaskGIT (CVPR 2022) which is a successful non-autoregressive Transformer based model compare with the proposed latent diffusion based method for code-tree generation?
- Adding more results highlighting the importance of using a latent diffusion model would be helpful to more strongly establish the benefits of the proposed method. Examples include comparing sampling time and parameter count, the effect of classifier free guidance for better conditioning, interpolating between topologically sketches by interpolating their code trees and checking for smoothness, etc.